# Cost efficient gradient boosting

**Sven Peter**
Heidelberg Collaboratory for Image Processing
Interdisciplinary Center for Scientific Computing
University of Heidelberg
69115 Heidelberg, Germany
sven.peter@iwr.uni-heidelberg.de

**Ferran Diego**
Robert Bosch GmbH
Robert-Bosch-Straße 200
31139 Hildesheim, Germany
ferran.diegoandilla@de.bosch.com

**Fred A. Hamprecht**
Heidelberg Collaboratory for Image Processing
Interdisciplinary Center for Scientific Computing
University of Heidelberg
69115 Heidelberg, Germany
fred.hamprecht@iwr.uni-heidelberg.de

**Boaz Nadler**
Department of Computer Science
Weizmann Institute of Science
Rehovot 76100, Israel
boaz.nadler@weizmann.ac.il

## Abstract

Many applications require learning classifiers or regressors that are both accurate *and* cheap to evaluate. Prediction cost can be drastically reduced if the learned predictor is constructed such that on the majority of the inputs, it uses cheap features and fast evaluations. The main challenge is to do so with little loss in accuracy. In this work we propose a budget-aware strategy based on deep boosted regression trees. In contrast to previous approaches to learning with cost penalties, our method can grow very deep trees that on average are nonetheless cheap to compute. We evaluate our method on a number of datasets and find that it outperforms the current state of the art by a large margin. Our algorithm is easy to implement and its learning time is comparable to that of the original gradient boosting. Source code is made available at http://github.com/svenpeter42/LightGBM-CEGB.

## 1 Introduction

Many applications need classifiers or regressors that are not only accurate, but also cheap to evaluate [33, 30]. Prediction cost usually consists of two different components: The acquisition or computation of the features used to predict the output, and the evaluation of the predictor itself. A common approach to construct an accurate predictor with low evaluation cost is to modify the classical empirical risk minimization objective, such that it includes a prediction cost penalty, and optimize this modified functional [33, 30, 23, 24].

In this work we also follow this general approach, and develop a budget-aware strategy based on deep boosted regression trees. Despite the recent re-emergence and popularity of neural networks, our choice of boosted regression trees is motivated by three observations:

(*i*) Given ample training data and computational resources, deep neural networks often give the most accurate results. However, standard feed-forward architectures route a single input component (for example, a single coefficient in the case of vectorial input) through most network units. While the computational cost can be mitigated by network compression or quantization [14], in the extreme case to binary activations only [16], the computational graph is fundamentally dense. In a standard decision tree, on the other hand, each sample is routed along a single path from the root to a leaf, thus

visiting typically only a small subset of all split nodes, the "units" of a decision tree. In the extreme case of a balanced binary tree, each sample visits only $\log(N)$ out of a total of $N$ nodes.

(*ii*) Individual decision trees and their ensembles, such as Random Forest [4] and Gradient Boosting [12], are still among the most useful and highly competitive methods in machine learning, particularly in the regime of limited training data, little training time and little expertise for parameter tuning [11].

(*iii*) When features and/or decisions come at a premium, it is convenient but wasteful to assume that all instances in a data set are created equal (even when assumed i.i.d.). Some instances may be easy to classify based on reading a single measurement / feature, while others may require a full battery of tests before a decision can be reached with confidence [35]. Decision trees naturally lend themselves to such a "sequential experimental design" setup: after first using cheap features to split all instances into subsets, the subsequent decisions can be based on more expensive features which are, however, only elicited if truly needed. Importantly, the set of more expensive features is requested *conditionally* on the values of features used earlier in the tree.

In this work we address the challenge of constructing an ensemble of trees that is both accurate and yet cheap to evaluate. We first describe the problem setup in Section 2, and discuss related work in Section 3. Our key contribution appears in Section 4, where we propose an extension of gradient boosting [12] which takes prediction time penalties into account. In contrast to previous approaches to learning with cost penalties, our method can grow very deep trees that on average are nonetheless cheap to compute. Our algorithm is easy to implement and its learning time is comparable to that of the original gradient boosting. As illustrated in Section 5, on a number of datasets our method outperforms the current state of the art by a large margin.

## 2   Problem setup

Consider a regression problem where the response $Y \in \mathbb{R}$ and each instance $X$ is represented by $M$ features, $X \in \mathbb{R}^M$. Let $L : \mathbb{R} \times \mathbb{R} \to \mathbb{R}$ be a loss function, and $\mathcal{T}$ be a set of admissible functions. In supervised learning, given a training set of $N$ pairs $(\boldsymbol{x_i}, y_i)$ sampled i.i.d. from $(X, Y)$, a classical approach to learn a predictor $T \in \mathcal{T}$ is to minimize the empirical loss $L$ on the training set,

$$\min_{T \in \mathcal{T}} \sum_{i=1}^{N} L(y_i, T(\boldsymbol{x}_i)). \tag{1}$$

In this paper we restrict ourselves to the set $\mathcal{T}$ that consists of an ensemble of trees, namely predictors of the form $T(\boldsymbol{x}) = \sum_{k=1}^{K} t_k(\boldsymbol{x})$. Each single decision tree $t_k$ can be represented as a collection of $L_k$ leaf nodes with corresponding responses $\boldsymbol{\omega}_k = (\omega_{k,1}, \ldots, \omega_{1,L_k}) \in \mathbb{R}^{L_k}$ and a function $q_k : \mathbb{R}^M \to \{1, \ldots, L_k\}$ that encodes the tree structure and maps an input to its corresponding terminal leaf index. The output of the tree is $t_k(\boldsymbol{x}) = \boldsymbol{\omega}_{k,q_k(\boldsymbol{x})}$.

Learning even a single tree that exactly minimizes the functional in Eq. (1) is NP-hard under several aspects of optimality [15, 19, 25, 36]. Yet, single trees and ensemble of trees are some of the most successful predictors in machine learning and there are multiple greedy based methods to construct tree ensembles that approximately solve Eq. (1) [4, 12, 11].

In many practical applications, however, it is important that the predictor $T$ is not only accurate but also fast to compute. Given a prediction cost function $\Psi : \mathcal{T} \times \mathbb{R}^M \to \mathbb{R}^+$ a standard approach is to add a penalty to the empirical risk minimization above [33, 30, 35, 23, 24]:

$$\min_{T \in \mathcal{T}} \sum_{i} L(y_i, T(\boldsymbol{x}_i)) + \lambda \Psi(T, \boldsymbol{x_i}). \tag{2}$$

The parameter $\lambda$ controls the tradeoff between accuracy and prediction cost.

Typically, the prediction cost function $\Psi$ consists of two components. The first is the cost of acquiring or computing relevant input features. For example, think of a patient at the emergency room where taking his temperature and blood oxygen levels are cheap, but a CT-scan is expensive. The second component is the cost of evaluating the function $T$, which in our case is the sum of the cost of evaluating the $K$ individual trees $t_k$.

In more detail, the first component of feature computation cost may also depend on the specific prediction problem. In some scenarios, test instances are independent of each other and the features

can be computed for each input instance on demand. But there are also others. In image processing, for example, where the input is an image which consists of many pixels and the task is to predict some function at all pixels. In such cases, even though specific features can be computed for each pixel independently, it may be cheaper or more efficient to compute the same feature, such as a separable convolution filter, at all pixels at once [1, 13]. The cost function $\Psi$ may be dominated in these cases by the second component - the time it takes to evaluate the trees.

After discussing related work in Section 3, in Section 4 we present a general adaptation of gradient boosting [12] to minimize Eq. (2), that takes into account both prediction cost components.

## 3  Related work

The problem of learning with prediction cost penalties has been extensively studied. One particular case is that of class imbalance, where one class is extremely rare and yet it is important to accurately annotate it. For example, the famous Viola-Jones cascades [31] use cheap features to discard examples belonging to the negative class. Later stages requiring expensive features are only used for the rare suspected positive class. While such an approach is very successful, due to its early exit strategy it cannot use expensive features for different inputs [20, 30, 9].

To overcome the limitations imposed by early exit strategies, various methods [34, 35, 18, 32] proposed single tree constructions but with more complicated decisions at the individual split nodes. The tree is first learned without taking prediction cost into account followed by an optimization step that includes this cost. Unfortunately, in practice these single-tree methods are inferior to current state-of-the-art algorithms that construct tree ensembles [23, 24].

BUDGETRF [23] is based on Random Forests and modifies the impurity function that decides which split to make, to take feature costs into account. BUDGETRF has several limitations: First, it assumes that tree evaluation cost is negligible compared to feature acquisition, and hence is not suitable for problems where features are cheap to compute and the prediction cost is dominated by predictor evaluation or were both components contribute equally. Second, during its training phase, each usage of a feature incurs its acquisition cost so repeated feature usage is not modeled, and the probability for reaching a node is not taken into account. At test time, in contrast, they do allow "free" reuse of expensive features and do compute the precise cost of reaching various tree branches. BUDGETRF thus typically does not yield deep but expensive branches which are only seldomly reached.

BUDGETPRUNE [24] is a pruning scheme for ensembles of decision trees. It aims to mitigate limitations of BUDGETRF by pruning expensive branches from the individual trees. An Integer Linear Program is formulated and efficiently solved to take repeated feature usage and probabilities for reaching different branches into account. This method results in a better tradeoff but still cannot create deep and expensive branches which are only seldomly reached if these were not present in the original ensemble. This method is considered to be state of the art when prediction cost is dominated by the feature acquisition cost [24]. We show in Section 5 that constructing deeper trees with our methods results in a significantly better performance.

GREEDYMISER [33], which is most similar to our work, is a stage-wise gradient-boosting type algorithm that also aims to minimize Eq. (2) using an ensemble of regression trees. When both prediction cost components are assumed equally significant, GREEDYMISER is considered state of the art. Yet, GREEDYMISER also has few limitations: First, all trees are assumed to have the *same* prediction cost for all inputs. Second, by design it constructs shallow trees all having the same depth. We instead consider individual costs for each leaf and thus allow construction of deeper trees. Our experiments in Section 5 suggest that constructing deeper trees with our proposed method significantly outperforms GREEDYMISER.

## 4  Gradient boosting with cost penalties

We build on the gradient boosting framework [12] and adapt it to allow optimization with cost penalties. First we briefly review the original algorithm. We then present our cost penalty in Section 4.1, the step wise optimization in 4.2 and finally our tree growing algorithm that builds trees with deep branches but low expected depth and feature cost in Section 4.3 (such a tree is shown in Figure 1b and compared to a shallow tree that is more expensive and less accurate in Figure 1a).

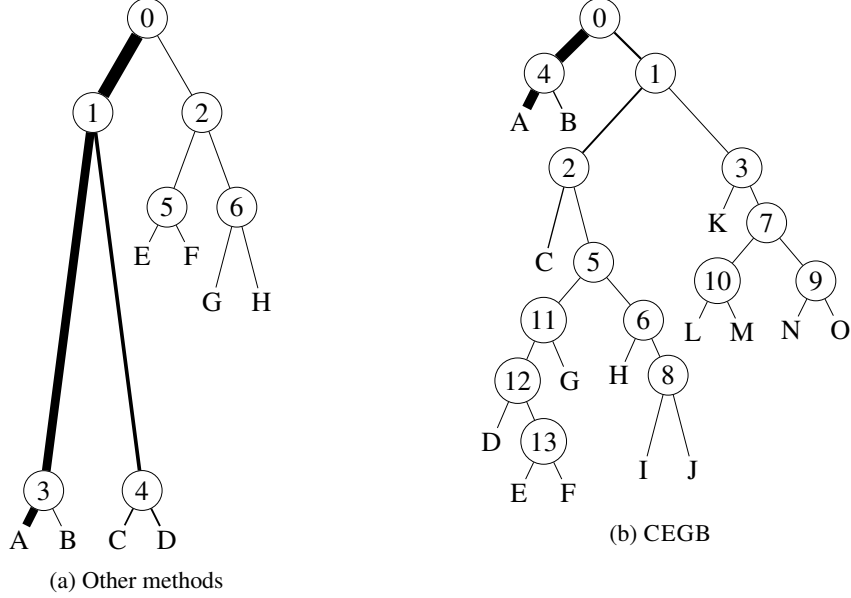

(a) Other methods

(b) CEGB

Figure 1: **Illustration of trees generated by the different methods**: Split nodes are numbered in the order they have been created, leaves are represented with letters. The vertical position of nodes corresponds to the feature cost required for each sample and the edge's thickness represents the number of samples moving along this edge. A tree constructed by GreedyMiser is shown in (a): The majority of samples travel along a path requiring a very expensive feature. BudgetPrune could only prune away leaves E,F,G and H which does not correspond to a large reduction in costs. CEGB however only uses two very cheap splits for almost all samples (leaves A and B) and builds a complex subtree for the minority that is hard to classify. The constructed tree shown in (b) is deep but nevertheless cheap to evaluate on average.

Gradient boosting tries to minimize the empirical risk of Eq. (1), by constructing a linear combination of $K$ weak predictors $t_k : \mathbb{R}^M \to \mathbb{R}$ from a set $\mathcal{F}$ of admissible functions (not necessarily decision trees). Starting with $T_0(\boldsymbol{x}) = 0$ each iteration $k > 0$ constructs a new weak function $t_k$ aiming to reduce the current loss. These boosting updates can be interpreted as approximations of the gradient descent direction in function space. We follow the notation of [8] who use gradient boosting with weak predictors $t_k$ from the set of regression trees $\mathcal{T}$ to minimize regularized empirical risk

$$\min_{t_1,\dots,t_K \in \mathcal{T}} \sum_{i=1}^{N} \left[ L(y_i, \sum_{k=1}^{K} t_k(\boldsymbol{x}_i)) \right] + \sum_{k=1}^{K} \Omega(t_k). \tag{3}$$

The regularization term $\Omega(t_k)$ penalizes the complexity of the regression tree functions. They assume that $\Omega(t_k)$ only depends on the number of leaves $L_k$ and leaf responses $\boldsymbol{w}_k$ and derive a simple algorithm to directly learn these. We instead use a more complicated prediction cost penalty $\Psi$ and use a different tree construction algorithm that allows optimization with cost penalties.

## 4.1 Prediction cost penalty

Recall that for each individual tree the prediction cost penalty $\Psi$ consists of two components: (i) the feature acquisition cost $\Psi_f$ and (ii) the tree evaluation cost $\Psi_{\text{ev}}$. However, this prediction cost for the $k$-th tree, which is fitted to the residual of all previous iterations, depends on the earlier trees. Specifically, for any input $\boldsymbol{x}$, features used in the trees of the previous iterations do not contribute to the cost penalty again. We thus use the indicator function $C : \mathbb{N}^0_{\leq K} \times \mathbb{N}_{\leq N} \times \mathbb{N}_{\leq M} \to \{0, 1\}$ with $C(k, i, m) = 1$ if and only if feature $m$ was used to predict $\boldsymbol{x}_i$ by any tree constructed prior to and including iteration $k$. Furthermore $\beta_m \geq 0$ is the cost for computing or acquiring feature $m$ for a single input $\boldsymbol{x}$. Then the feature cost contribution $\Psi_{\text{f}} : \mathbb{N}^0_{\leq K} \times \mathbb{N}_{\leq N} \to \mathbb{R}_+$ of $\boldsymbol{x}_i$ for the first $k$ trees

is calculated as

$$\Psi_{\mathrm{f}}(k,i) = \sum_{m=1}^{M} \beta_m C(k,i,m) \tag{4}$$

Features computed for all inputs at once (e.g. separable convolution filters) contribute to the penalty independent of the instance $\boldsymbol{x}$ being evaluated. For those we use $\gamma_m$ as their total computation cost and define the indicator function $D : \mathbb{N}^0_{\leq K} \times \mathbb{N}_{\leq M} \to \{0,1\}$ with $D(k,m) = 1$ if and only if feature $m$ was used for any input $\boldsymbol{x}$ in any tree constructed prior to and including iteration $k$. Then

$$\Psi_{\mathrm{c}}(k) = \sum_{m=1}^{M} \gamma_m D(k,m) \tag{5}$$

The evaluation cost $\Psi_{\mathrm{ev},k} : \mathbb{N}_{\leq L_k} \to \mathbb{R}_+$ for a single input $\boldsymbol{x}$ passing through a tree is the number of split nodes between the root node and the input's terminal leaf $q_k(\boldsymbol{x})$, multiplied by a suitable constant $\alpha \geq 0$ which captures the cost to evaluate a single split. The total cost $\Psi_{\mathrm{ev}} : \mathbb{N}^0_{\leq K} \times \mathbb{N}_{\leq N} \to \mathbb{R}_+$ for the first $k$ trees is the sum of the costs of each tree

$$\Psi_{\mathrm{ev}}(k,i) = \sum_{\tilde{k}=1}^{k} \Psi_{\mathrm{ev},\tilde{k}}(q_{\tilde{k}}(\boldsymbol{x}_i)). \tag{6}$$

## 4.2 Tree Boosting with Prediction Costs

We have now defined all components of Eq. (2). Simultaneous optimization of all trees $t_k$ is intractable. Instead, as in gradient boosting , we minimize the objective by starting with $T_0(\boldsymbol{x}) = 0$ and iteratively adding a new tree at each iteration.

At iteration $k$ we construct the $k$-th regression tree $t_k$ by minimizing the following objective

$$O_k = \sum_{i=1}^{N} \left[ L(y_i, T_{k-1}(\boldsymbol{x}_i) + t_k(\boldsymbol{x}_i)) + \lambda \Psi(k, \boldsymbol{x}_i) \right] + \lambda \Psi_{\mathrm{c}}(k) \tag{7}$$

with $\Psi(k, \boldsymbol{x}_i) = \Psi_{\mathrm{ev}}(k,i) + \Psi_{\mathrm{f}}(k,i)$. Note that the penalty for features, which are computed for all inputs at once, $\Psi_{\mathrm{c}}(k)$ does not depend on $\boldsymbol{x}$ but only on the structure of the current and previous trees.

Directly optimizing the objective $O_k$ w.r.t. the tree $t_k$ is difficult since the argument $t_k$ appears inside the loss function. Following [8] we use a second order Taylor expansion of the loss around $T_{k-1}(\boldsymbol{x}_i)$. Removing constant terms from earlier iterations the objective function can be approximated by

$$O_k \approx \tilde{O}_k = \sum_{i=1}^{N} \left[ g_i t_k(\boldsymbol{x_i}) + \frac{1}{2} h_i t_k^2(\boldsymbol{x_i}) + \lambda \Delta \Psi(\boldsymbol{x_i}) \right] + \lambda \Delta \Psi_{\mathrm{c}} \tag{8}$$

where

$$g_i = \partial_{\hat{y}_i} L(y_i, \hat{y}_i) \Big|_{\hat{y}_i = T_{k-1}(\boldsymbol{x_i})}, \tag{9a} \qquad h_i = \partial^2_{\hat{y}_i} L(y_i, \hat{y}_i) \Big|_{\hat{y}_i = T_{k-1}(\boldsymbol{x_i})}, \tag{9b}$$

$$\Delta \Psi(\boldsymbol{x}_i) = \Psi(k, \boldsymbol{x}_i) - \Psi(k-1, \boldsymbol{x}_i), \tag{9c} \qquad \Delta \Psi_{\mathrm{c}} = \Psi_{\mathrm{c}}(k) - \Psi_{\mathrm{c}}(k-1). \tag{9d}$$

As in [8] we rewrite Eq. (8) for a decision tree $t_k(\boldsymbol{x}) = \boldsymbol{\omega}_{k, q_k(\boldsymbol{x})}$ with a fixed structure $q_k$,

$$\tilde{O}_k = \sum_{l}^{L_k} \left[ \left( \sum_{i \in \mathcal{I}_l} g_i \right) \boldsymbol{\omega}_{k,l} + \frac{1}{2} \left( \sum_{i \in \mathcal{I}_l} h_i \right) \boldsymbol{\omega}^2_{k,l} + \lambda \left( \sum_{i \in \mathcal{I}_l} \Delta \Psi(\boldsymbol{x}_i) \right) \right] + \lambda \Delta \Psi_{\mathrm{c}} \tag{10}$$

with the set $\mathcal{I}_l = \{i | q_k(\boldsymbol{x}_i) = l\}$ containing inputs in leaf $l$. For this fixed structure the optimal weights and the corresponding best objective reduction can be calculated explicitly:

$$\boldsymbol{\omega}^*_{k,l} = -\frac{\sum_{i \in \mathcal{I}_l} g_i}{\sum_{i \in \mathcal{I}_l} h_i}, \quad \text{(11a)} \quad \tilde{O}^*_k = -\frac{1}{2} \sum_{l}^{L} \left[ \frac{(\sum_{i \in \mathcal{I}_l} g_i)^2}{\sum_{i \in \mathcal{I}_l} h_i} + \lambda \left( \sum_{i \in \mathcal{I}_l} \Delta \Psi(\boldsymbol{x}_i) \right) \right] + \lambda \Delta \Psi_{\mathrm{c}} \quad \text{(11b)}$$

As we shall see in the next section, our cost-aware impurity function depends on the difference of Eq. (10) which results by replacing a terminal leaf with a split node [8]. Let $p$ be any leaf of the tree that can be converted to a split node and two new children $r$ and $l$ then the difference of Eq. (10) evaluated for the original and the modified tree is

$$\Delta \tilde{O}_k^{\text{split}} = \frac{1}{2} \left[ \frac{(\sum_{i \in \mathcal{I}_r} g_i)^2}{\sum_{i \in \mathcal{I}_r} h_i} + \frac{(\sum_{i \in \mathcal{I}_l} g_i)^2}{\sum_{i \in \mathcal{I}_l} h_i} - \frac{(\sum_{i \in \mathcal{I}_p} g_i)^2}{\sum_{i \in \mathcal{I}_p} h_i} \right] - \lambda \, \Delta \Psi_k^{\text{split}} \tag{12}$$

Let $m$ be the feature used by the node $s$ that we are considering to split. Then

$$\Delta \Psi_k^{\text{split}} = \underbrace{|\mathcal{I}_p|\alpha}_{\Psi_{\text{ev},k}^{\text{split}}} + \gamma_m \overbrace{\left(1 - D(k,m)\right)}^{\substack{\text{is feature } m \\ \text{used for the first time?}}} + \underbrace{\sum_{i \in \mathcal{I}_p} \beta_m \overbrace{\left(1 - C(k,i,m)\right)}^{\substack{\text{is feature } m \\ \text{of instance } \boldsymbol{x}_i \text{ used for the first time?}}}}_{\Psi_{\text{f},k}^{\text{split}}} \tag{13}$$

### 4.3   Learning a weak regressor with cost penalties

With these preparations we can now construct the regression trees. As mentioned above, this is a NP-hard problem. We use a greedy algorithm to grow a tree that approximately minimizes Eq. (10).

Standard algorithms that grow trees start from a single leaf containing all inputs. The tree is then iteratively expanded by replacing a single leaf with a split node and two new child leaves [4]. Typically this expansion happens in a predefined leaf order (breadth- or depth-first). Splits are only evaluated *locally* at a single leaf to select the best feature. The expansion is stopped once leaves are pure or once a maximum depth has been reached. Here, in contrast, we adopt the approach of [29] and grow the tree in a best-first order. Splits are evaluated for *all* current leaves and the one with the best objection reduction according to Eq. (12) is chosen. The tree can thus grow at any location. This allows to compare splits across different leaves and features at the same time (figure 1b shows an example for a best-first tree while figure 1a shows a tree constructed in breadth-first order). Instead of limiting the depth we limit the number of leaves in each tree to prevent over fitting.

This procedure has an important advantage when optimizing with cost penalties: Growing in a predefined order usually leads to balanced trees - all branches are grown independent of the cost. Deep and expensive branches using only a tiny subset of inputs are not easily possible. In contrast, growing at the leaf that promises the best tradeoff as given by Eq. (12) encourages growth on branches that contain few instances or growth using cheap features. Growth on branches that contain many instances or growth that requires expensive features is penalized. This strategy results in deep trees that are nevertheless cheap to compute on average. Figure 1 compares an individual tree constructed by others methods to the deeper tree constructed by CEGB.

We briefly compare our proposed strategy to GREEDYMISER: When we limit Eq. (8) to first order terms only, use breadth-first instead of best-first growth, assume that features always have to be computed for all instances at once and limit the tree depth to four we minimize Eq. (18) from [33]. GreedyMiser can therefore be represented as a special case of our proposed algorithm.

## 5   Experiments

The Yahoo! Learning to Rank (Yahoo! LTR) challenge dataset [7] consists of 473134 training, 71083 validation and 165660 test document-query pairs with labels $\{0, 1, 2, 3, 4\}$ where 0 means the document is irrelevant and 4 that it is highly relevant to the query. Computation cost for the 519 features used in the dataset are provided [33] and take the values $\{1, 5, 10, 20, 50, 100, 150, 200\}$. Prediction performance is evaluated using the Average Precision@5 metric which only considers the five most relevant documents returned for a query by the regressor [33, 23, 24]. We use the dataset provided by [7] and used in [23, 24].

We consider two different settings for our experiments, (*i*) feature acquisition and classifier evaluation time both contribute to prediction cost and (*ii*) classifier evaluation time is negligible w.r.t feature acquisition cost.

The first setting is used by GREEDYMISER. Regression trees with depth four are constructed and assumed to approximately cost as much as features with feature cost $\beta_m = 1$. We therefore set the

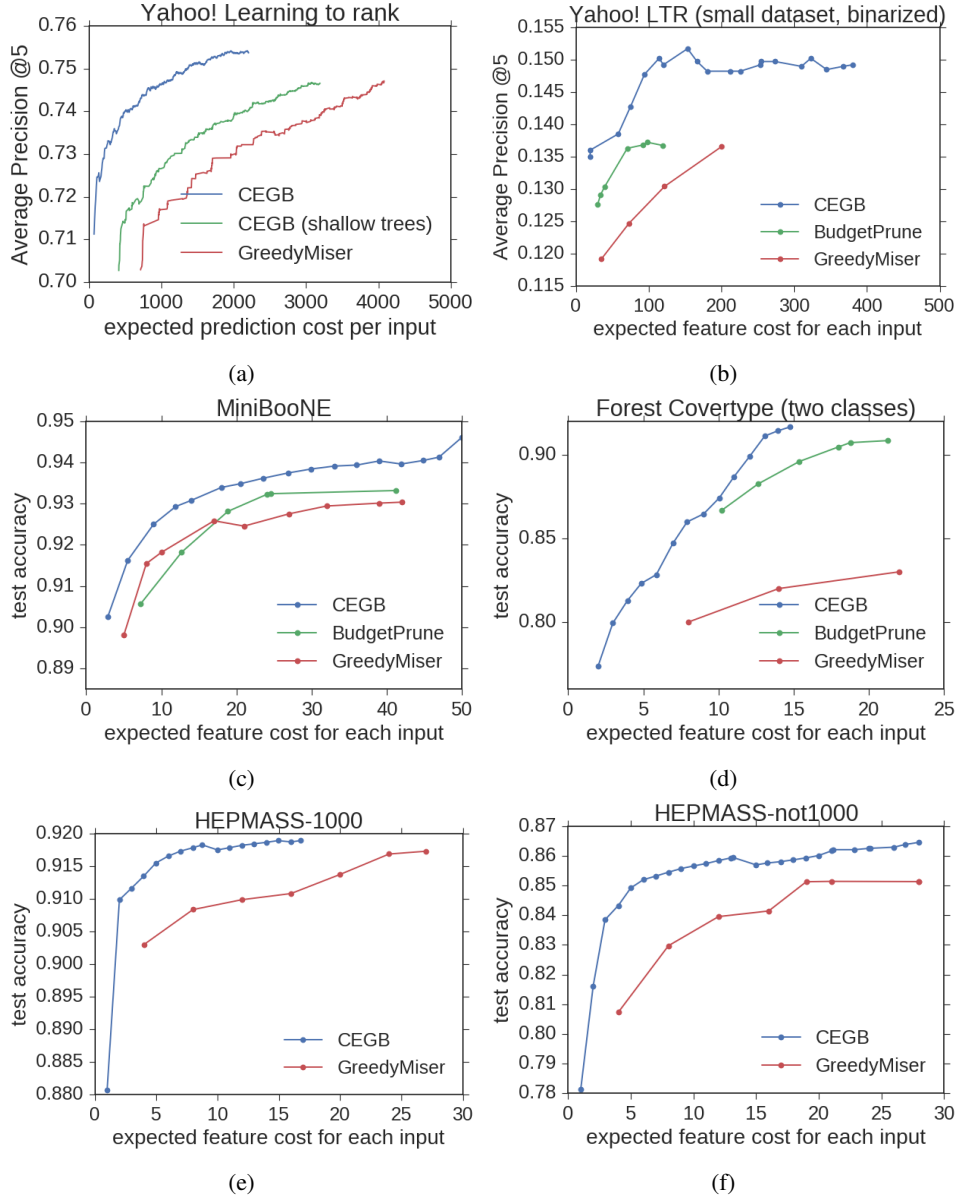

Figure 2: **Comparison against state of the art algorithms**: The Yahoo! LTR dataset has been used for (2a) and (2b) in different settings. In (2a) both tree evaluation and feature acquisition cost is considered. In (2b) only feature acquisition cost is shown. (2c) shows results on the MiniBooNE dataset with uniform feature costs. GREEDYMISER and BUDGETPRUNE results for (2b), (2c) and (2d) from [24]. BUDGETPRUNE did not finish training on the HEPMASSS datasets to due their size and the associated CPU time and RAM requirements. CEGB is our proposed method.

split cost $\alpha = \frac{1}{4}$ to allow a fair comparison with our trees which will contain deeper branches. We also use our algorithm to construct trees similar to GREEDYMISER by limiting the trees to 16 leaves with a maximum branch depth of four. Figure 2a shows that even the shallow trees are already always strictly better than GREEDYMISER. This happens because our algorithm correctly accounts for the different probabilities of reaching different leaves (see also figure 1). When we allow deep branches the proposed method gives significantly better results than GREEDYMISER and learns a predictor with better accuracy at a much lower cost.

The second setting is considered by BUDGETPRUNE. It assumes that feature computation is much more expensive than classifier evaluation. We set $\alpha = 0$ to adapt our algorithm to this setting.

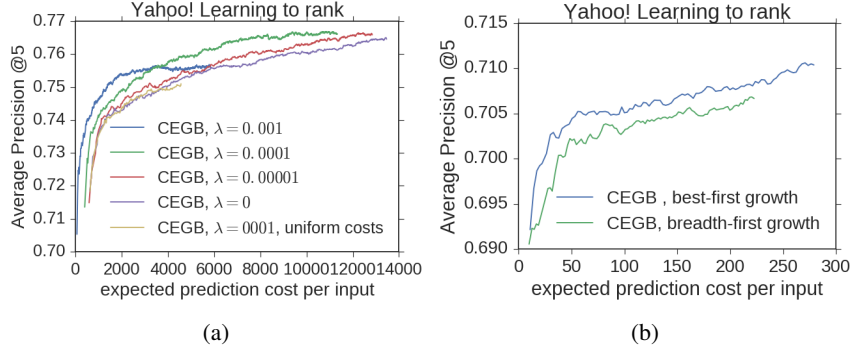

(a)                                                                 (b)

Figure 3: In (3a) we study the influence of the feature penalty on the learned classifier. (3b) shows how best-first training results in better precision given the same cost budget.

The dataset is additionally binarized by setting all targets $y > 0$ to $y = 1$. GREEDYMISER has a disadvantage in this setting since it works on the assumption that the cost of each tree is independent of the input $x$. We still include it in our comparison as a baseline. Figure 3b shows that our proposed method again performs significantly better than others. This confirms that we learn a classifier with very expected cheap prediction cost in terms of both feature acquisition and classifier evaluation time.

The MiniBooNE dataset [27, 21] consists of 45523 training, 19510 validation and 65031 test instances with labels $\{0, 1\}$ and 50 features. The Forest Covertype dataset [3, 21] consists of 36603 training, 15688 validation and 58101 test instances with 54 features restricted to two classes as done in [24]. Feature costs are not available for either dataset and assumed to be uniform, i.e. $\beta_m = 1$. Since no relation between classifier evaluation and feature cost is known we only compute the latter to allow a fair comparison, as in [24]. Figure 2c and 2d show that our proposed method again results in a significantly better predictor than both GREEDYMISER and BUDGETPRUNE.

We additionally use the HEPMASS-1000 and HEPMASS-not1000 datasets [2, 21]. Similar to MiniBooNE no feature costs are known and we again uniformly set them to one for all features, i.e. $\beta_m = 1$. Both datasets contain over ten million instances which we split into 3.5 million training, 1.4 million validation and 5.6 million test instances. These datasets are much larger than the others and we did not manage to successfully run BUDGETPRUNE due to its RAM and CPU time requirements. We only report results on GREEDYMISER and our algorithm in Figure 2e and 2f. CEGB again results in a classifier with a better tradeoff than GREEDYMISER.

## 5.1 Influence of feature cost and tradeoff parameters

We use the Yahoo! LTR dataset to study the influence of the features costs $\beta$ and the tradeoff parameter $\lambda$ on the learned regressor. Figure 3a shows that regressors learned with a large $\lambda$ reach similar accuracy as those with smaller $\lambda$ at a much cheaper cost. Only $\lambda = 0.001$ converges to a lower accuracy while others approximately reach the same final accuracy. The tradeoff is shifted towards using cheap features too strongly. Such a regressor is nevertheless useful when the problems requires very cheap results and the final improvement in accuracy does not matter.

Next, we set all $\beta_m = 1$ during training time only and use the original cost during test time. The learned regressor behaves similar to one learned with $\lambda = 0$. This shows that the regressors save most of the cost by limiting usage of expensive features to a small subset of inputs.

Finally we compare breadth-first to best-first training in Figure 3b. We use the same number of leaves and trees and try to build a classifier that is as cheap as possible. Best-first training always reaches a higher accuracy for a given prediction cost budget. This supports our observation that deep trees which are cheap to evaluate on average are important for constructing cheap and accurate predictors.

## 5.2 Multi-scale classification / tree structure optimization

In images processing, classification using multiple scales has been extensively studied and used to build fast or more more accurate classifiers [6, 31, 10, 26]. The basic idea of these schemes is

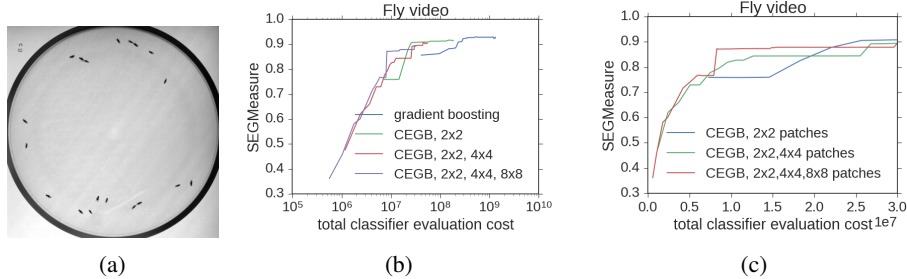

(a)          (b)          (c)

Figure 4: **Multi-scale classification**: (4a) shows a single frame from the dataset we used. (4b) shows how our proposed algorithm CEGB is able to build significantly cheaper trees than normal gradient boosting. (4c) zooms into the region showing the differences between the various patch sizes.

that a large image is downsampled to increasingly coarse resolutions. A multi-scale classifier first analyzes the coarsest resolution and decides whether a pixel on the coarse level represents a block of homogeneous pixels on the original resolution, or if analysis on a less coarse resolution is required. Efficiency results from the ability to label many pixels on the original resolution at once by labeling a single pixel on a coarser image.

We use this setting as an example to show how our algorithm is also capable of optimizing problems where feature cost is negligible compare to predictor evaluation cost. Inspired by average pooling layers in neural networks [28] and image pyramids [5] we first compute the average pixel values across non-overlapping 2x2, 4x4 and 8x8 blocks of the original image. We compute several commonly used and very fast convolutional filters on each of those resolutions. We then replicated these features values on the original resolution, e.g. the feature response of a single pixel on the 8x8-averaged image is used for all 64 pixels We modify Eq. (12) and set $\Delta\Psi_k^{\text{split}} = |\mathcal{I}_p|\alpha\epsilon_m$ where $\epsilon_m$ is the number of pixels that share this feature value, e.g. $\epsilon_m = 64$ when feature $m$ was computed on the coarse 8x8-averaged image.

We use forty frames with a resolution of 1024x1024 pixels taken from a video studying fly ethology. Our goal here is to detect flies as quickly as possible, as preprocessing for subsequent tracking. A single frame is shown in Figure 4a. We use twenty of those for training and twenty for evaluation. Accuracy is evaluated using the SEGMeasure score as defined in [22]. Comparison is done against regular gradient boosting by setting $\Psi = 0$.

Figure 4b shows that our algorithm constructs an ensemble that is able to reach similar accuracy with a significantly smaller evaluation cost. Figure 4c shows more clearly how the different available resolutions influence the learned ensemble. Coarser resolutions allow a very efficient prediction at the cost of accuracy. Overall these experiments show that our algorithm is also capable of learning predictors that are cheap while maintaining accuracy even when the evaluation cost of these dominates w.r.t the feature acquisition cost.

## 6  Conclusion

We presented an adaptation of gradient boosting that includes prediction cost penalties, and devised fast methods to learn an ensemble of deep regression trees. A key feature of our approach is its ability to construct deep trees that are nevertheless cheap to evaluate on average. In the experimental part we demonstrated that this approach is capable of handing various different settings of prediction cost penalties consisting of feature cost and tree evaluation cost. Specifically, our method significantly outperformed state of the art algorithms GREEDYMISER and BUDGETPRUNE when feature cost either dominates or contributes equally to the total cost.

We additionally showed an example where we are able to optimize the decision structure of the trees itself when evaluation of these is the limiting factor.

Our algorithm can be easily implemented using any gradient boosting library and does not slow down training significantly. For these reasons we believe it will be highly valuable for many applications. Source code based on LightGBM [17] is available at http://github.com/svenpeter42/LightGBM-CEGB.

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
