[Supplementary Material]

# Supplementary material for the paper:
# Cost efficient gradient boosting

**Sven Peter**
Heidelberg Collaboratory for Image Processing
Interdisciplinary Center for Scientific Computing
University of Heidelberg
69115 Heidelberg, Germany
sven.peter@iwr.uni-heidelberg.de

**Ferran Diego**
Robert Bosch GmbH
Robert-Bosch-Straße 200
31139 Hildesheim, Germany
ferran.diegoandilla@de.bosch.com

**Fred A. Hamprecht**
Heidelberg Collaboratory for Image Processing
Interdisciplinary Center for Scientific Computing
University of Heidelberg
69115 Heidelberg, Germany
fred.hamprecht@iwr.uni-heidelberg.de

**Boaz Nadler**
Department of Computer Science
Weizmann Institute of Science
Rehovot 76100, Israel
boaz.nadler@weizmann.ac.il

## 8   GreedyMiser

As mentioned in the main paper GREEDYMISER can be viewed as a special case of our algorithm. We repeat our objective function (Eq. (8) in the main paper):

$$O_k \approx \tilde{O}_k = \sum_{i=1}^{N} \left[ g_i t_k(\boldsymbol{x_i}) + \frac{1}{2} h_i t_k^2(\boldsymbol{x_i}) + \lambda \Delta \Psi(\boldsymbol{x_i}) \right] + \Delta \Psi_{\mathrm{c}} \tag{1}$$

where

$$g_i = \partial_{\hat{y}_i} L(y_i, \hat{y}_i) \Big|_{\hat{y}_i = T_{k-1}(\boldsymbol{x_i})}, \tag{2a}$$

$$h_i = \partial_{\hat{y}_i}^2 L(y_i, \hat{y}_i) \Big|_{\hat{y}_i = T_{k-1}(\boldsymbol{x_i})}, \tag{2b}$$

$$\Delta \Psi(\boldsymbol{x_i}) = \Psi(k, \boldsymbol{x_i}) - \Psi(k-1, \boldsymbol{x_i}), \tag{2c}$$

$$\Delta \Psi_{\mathrm{c}} = \Psi_{\mathrm{c}}(k) - \Psi_{\mathrm{c}}(k-1). \tag{2d}$$

GreedyMiser's objective is given as Eq. (18) in [1]:

$$\sum_{i=1}^{n} -r_i h_t(\boldsymbol{x_i}) + \frac{\lambda}{\eta} \sum_{\alpha} c_\alpha \psi_\alpha F_{\alpha t} \tag{3}$$

The different notations are shown in table 1. When the following restrictions are used in CEGB we solve the same optimization problem as GREEDYMISER:

- Use only first order derivatives of the loss function (i.e. $h_i = 0$).
- Limit trees to depth four and assume that the tree evaluation cost for all samples is constant instead of taking into account the probabilities for reaching the different leaves.
- Assume that all features always have to computed for all samples at once instead of allowing lazy computation in the split nodes.
- Use breadth-first instead of best-first tree growth.

Table 1: CEGB and GreedyMiser notation for constructing a new tree

|  | GREEDYMISER | CEGB |
|---|---|---|
| boosting iteration | t | k |
| regression tree response | $h_t(\boldsymbol{x})$ | $t_k(\boldsymbol{x})$ |
| first order derivative | $-r_i$ | $g_i$ |
| second order derivative | 0 | $h_i$ |
| feature index | $\alpha$ | m |
| additional cost required by the next tree for features computed for all samples at once | $c_\alpha \phi_\alpha F_{\alpha t}$ | $\Delta \Psi_c$ |
| additional cost required by the next tree for features computed for single samples on demand | not part of the model | $\Delta \Psi_f$ |
| additional cost required by the next tree for evaluating the tree itself | $e$ (usually set to 1) | $\alpha \sum_{l=0}^{L_k} |\mathcal{I}_l|$ |

## 9 Trees constructed for the Yahoo LTR dataset

In this section we will show some examples for trees constructed by CEGB and GREEDYMISER for the experiments shown in figure (2b) done using the Yahoo Learning to Rank dataset.

GREEDYMISER does not take the probabilties for reaching different leaves into account and therefore is only capable of learning shallow trees. Even in those expensive features are used near the root node for almost all samples.

CEGB is capable of learning deep trees that are nevertheless cheap to evaluate on average instead by using a different cost penalty and making use of best-first growth. Many samples are classified using only one or two very cheap splits while a small subset of samples reaches much deeper branches which occasionally make use of the more expensive features.

Figure 1: **Different trees learned by GREEDYMISER**: The number inside the split node denotes the feature index. The vertical position of nodes corresponds to the feature cost required for each sample and the edge's thickness represents the number of samples moving along this edge. The first tree already uses feature 188 with a cost of $\beta_{188} = 20$ for a majority of all samples.

Figure 2: **Different trees learned by CEGB**: The number inside the split node denots the feature index. The vertical position of nodes corresponds to the feature cost required for each sample and the edge's thickness represents the number of samples moving along this edge. The different to the trees learned by GREEDYMISER is easy to see: The majority of samples require at most two splits until they reach their final leaf. The minority is sent through a much more complex subtree where expensive features can be used. On average this results in a tree that is still cheap to evaluate but has a much better predictve performance.

(a) Penalty for each split    (b) Breadth first tree growth    (c) Depth first tree growth    (d) **Best-first tree growth**

Figure 3: **Tree growth strategies**: Split nodes are shown as circles, leaves as squares. For (3b) to (3d) the number in each node indicates at which iteration it has been created. Assume that the split nodes marked in red result in a bad tradeoff (high cost for moderate loss reduction) and the split node marked in green result in a very good tradeoff (low cost for large loss reduction). When using breadth-first (3b) or depth-first (3c) growth the bad split is created before the good split. When using leaf-first growth in (3d) however the bad split is created at the very end because of the high cost penalty while the good split is created directly after its parent node. Instead of limiting the depth, the number of leaves can be limited - this strategy grows deep trees that are nevertheless cheap to compute on average.

## 10 Algorithm

In addition to the explanation given in the main paper we also show pseudo code for the construction of individual trees and learning of the ensemble.

---

**Algorithm 1** CEGB

---

**Require:** $(x_i, y_i), \alpha, \boldsymbol{\beta}, \boldsymbol{\gamma}, K \in \mathbb{N}_{>0}$
**Initialize:** $\tilde{C} \in \{0,1\}^{N \times M}, \tilde{C}_{im} = 1$
**Initialize:** $D \in \{0,1\}^M, D_m = 1$
 1: $T_0 \leftarrow 0$
 2: **for** $k = 1$ **to** $K$ **do**
 3: $\quad g_i = \partial_{\hat{y}_i} L(y_i, \hat{y}_i)\Big|_{\hat{y}_i = T_{k-1}(\boldsymbol{x_i})} \quad \forall\, i$
 4: $\quad h_i = \partial^2_{\hat{y}_i} L(y_i, \hat{y}_i)\Big|_{\hat{y}_i = T_{k-1}(\boldsymbol{x_i})} \quad \forall\, i$
 5: $\quad t_k \leftarrow$ Result of Alg. (2)
 6: $\quad T_k \leftarrow T_{k-1} + t_k$
 7: **end for**
 8: **return** $T_K$

---

---

**Algorithm 2** BEST-FIRST TREE LEARNING

---

**Require:** $(x_i, g_i, h_i), \boldsymbol{\beta}, \boldsymbol{\gamma}, \tilde{C}, D$
**Initialize:** a single leaf with all $(x_i, g_i, h_i)$
 1: **for** 1 **to** max_leaves **do**
 2: $\quad$ Find best (split $s$, feature $m$, leaf $p$) using Eq. (13) with $\Psi^{\text{split}}_{\text{f},k} = \gamma_m D_m + \sum_{i \in \mathcal{I}_p} \beta_m \tilde{C}_{im}$ for all terminal leaves
 3: $\quad$ Convert $p$ to a split node using split $s$
 4: $\quad$ Add two new children to $p$
 5: $\quad$ Set $\tilde{C}_{im} = 0 \; \forall i \in \mathcal{I}_p$
 6: $\quad$ Set $D_m = 0$
 7: **end for**
 8: **return** Tree

---

# References

[1] Zhixiang Xu, Kilian Weinberger, and Olivier Chapelle. The greedy miser: Learning under test-time budgets. In John Langford and Joelle Pineau, editors, *Proceedings of the 29$^{th}$ International Conference on Machine Learning (ICML-12)*, ICML '12, pages 1175–1182, July 2012.