[Reviews · NeurIPS 2017]

Reviewer 1



1. Summary of paper This paper introduced a technique to learn a gradient boosted regression tree ensemble that is sensitive to feature costs and the cost of evaluating splits in the tree. Thus, the paper is similar to the work of Xu et al., 2012. The main differences are the fact that the feature and evaluation costs are input-specific, the evaluation cost depends on the number of tree splits, their optimization approach is different (based on the Taylor expansion around T_{k-1}, as described in the XGBoost paper), and they use best-first growth to grow the trees to a maximum number of splits (instead of a max depth). The authors point out that their setup works either in the case where feature cost dominates or evaluation cost dominates and they show experimental results for these settings. 2. High level subjective The paper is clearly written. Apart from introducing a significant amount of notation, I find it clear to follow. The contribution of the paper seems a bit small given the work of XGBoost and GreedyMiser. The paper seems to have sufficient experiments comparing the method with prior work and showing how model parameters affect performance. I have one confusion which may warrant an additional experiment, which I describe below. I think this model could be interesting for practitioners interested in classification in time-constrained settings. 3. High level technical One thing that is unclear is in Figures 2a and 2b, how was the cost measured for each method to obtain the Precision versus Cost curves? It seems to me that for CEGB cost is measured per-input and per-split, but for the other methods cost is measured per-tree. If this is the case this seems a bit unfair. It seems to me that the cost of each method should be evaluated in exactly the same way in order to compare each method fairly, even if the original papers evaluated cost in a different way. For this reason it's unclear to me how much better CEGB is than the other methods. If the costs are measured differently I think it would be really helpful to modify Figure 2 so that all of the costs are measured the same way. Personally I think the paper would improve if Figure 1 was removed. I think it's pretty clear what you mean by breadth-first vs. depth-first vs. best-first. I would replace this figure with one that shows the trees generated by CEGB, GreedyMiser and BudgetPrune for the Yahoo! Learning to Rank dataset. And it shows in detail the features and the costs at each split. Something that gives the reader a bit more insight as to why CEGB is outperforming the other methods. I think it would also be nice to more clearly emphasize what improvements over GreedyMiser you make to arrive at CEGB. For instance, it would be nice to write out GreedyMiser in the same notation and show the innovations (best-first, not limited-depth, different optimization procedure) that lead to CEGB. Then you could have a figure showing how each innovation improves the Precision vs. Cost curve. In Section 5.2 it's not completely clear to me why GreedyMiser or BudgetPrune cannot be applied here. Why is the cost of feature responses considered an evaluation cost and not a feature cost? 4. Low level technical - In equation 7, why is \lambda only in front of the first cost term and not the second? - In equation 12, should the denominators of the first 3 terms have a term with \lambda added, similar to the XGBoost paper? - What do the 'levels' refer to in Figure 4b? - Line 205: I would recommend redefining \beta_m here as readers may forget its definition. Similarly with \alpha on Line 213. 5. Summary of review While this paper aims to introduce a new method for cost efficient learning that (a) outperforms the state-of-the-art and (b) works in settings where previous methods are unsuitable, it is unclear to me if these results are because (a) the cost is measured differently and (b) because a cost is defined as an evaluation cost instead of a feature cost. Until these confusions are resolved I think the paper is slightly below the acceptance threshold of NIPS.

Reviewer 2



**summary: The paper proposes a gradient boosting approach (CEGB) for prediction under budget constraints. During training CEGB minimizes empirical loss plus a cost penalty term. Three types of costs are incorporated: feature cost per example and across all examples as well as evaluation cost. It then looks for the best split greedily that minimizes the second order approximation of the loss objective. Experiments on show improved performance over state of the art methods on two datasets. **Significance: The proposed algorithm is closely related to GreedyMiser (Xu et al 2012). The main difference lies in the construction of the weak regressors. The key contribution of CEGB is to allow new splits at any current leaves rather than in a pre-defined fashion. To alleviate the computation for searching for the best leaf to split in each iteration, second-order approximation of the loss is used. Although the ideas of best-first tree learning, second-order loss approximation are well explored in the gradient boosting literature, applying them to the cost efficient setting is novel. CEGB is thus a refinement of GreedyMiser. **Results: CEGB is only compared to the state-of-the-art methods BudgetPrune and GreedyMiser on two datasets: Yahoo and MiniBooNE. In particular, the authors did not include the Forest dataset as used by BudgetPrune (Nan et al 2016). My experience is that gradient boosting based methods perform poorly compared to random forests based methods (BudgetPrune) on the Forest dataset. This paper would be made stronger if more datasets are included to compare different methods. On another note, the results in Figure 2(b) is exceedingly good for CEGB without tree cost. It's quite surprising (and hard to believe) that the precision can rise above 0.14 with average feature cost of ~20. How many trees are used and what's the average number of nodes in each tree to get such a good performance? **Others: The authors repeatedly claimed that BudgetPrune assumes that feature computation is much more expensive than evaluation cost and therefore irrelevant in several experiments. This is not true. The formulation of of BudgetPrune by Nan et al 2016 does take into account evaluation costs and it can be traded-off with the feature cost. **Overall: I think this paper provides a refinement of GreedyMiser and the experimental results look promising. Adding more datasets and comparisons would make it even stronger. **After discussion with the reviewers, I feel that the novelty of this paper is not that much given the work of XGBoost and GreedyMiser. But the experimental results show improvement over the state-of-the-art. The authors promised additional datasets to be included in the final version. So I'm still leaning to accept this paper.

Reviewer 3



The authors present a novel algorithm to learn ensembles of deep decision (or regression) trees that optimize both prediction accuracy and feature cost. The resulting prediction system is adaptive in the sense that for different inputs different features sets of features will be acquired/evaluated thus resulting in a reduced cost while mainitaing prediction accuracy. The authors approach is an iterative algorithm that extends gradient tree boosting with a cost-sentive objective function. The algorithm iteretavely adds decision trees to the ensemble. Each new candidate tree is trained to approximately optimize the cost of the entire ensemble. To accomplish individual tree learning given the cost-constrains w.r.t the existing ensemble, the authors introduce a cost-aware impurity function. The authors demonstrate that their algorithm outperforms existing methods in cost-aware ensemble learning on several real datasets. Overall, I think the paper is well written, the algorithm derivations are interesting and novel, and the problem being addressed is highly relevant to many domains.